# c-Src functionality controls self-renewal and glucose metabolism in MCF7 breast cancer stem cells

Víctor Mayoral-Varo[1☉], Annarica Calcabrini[1☉¤a], María Pilar Sánchez-Bailón[1☉¤b], Óscar H. Martínez-Costa[1], Cristina González-Páramos[1], Sergio Ciordia[2], David Hardisson[3,4,5], Juan J. Aragón[1], Miguel Ángel Fernández-Moreno[1,6,7], Jorge Martín-Pérez[1,5]*

1 Instituto de Investigaciones Biomédicas A. Sols (CSIC/UAM), Madrid, Spain, 2 Servicio de Espectrometría de Masas, Centro Nacional de Biotecnología (CSIC), Madrid, Spain, 3 Servicio de Anatomía Patológica, Hospital Universitario La Paz, Madrid, 4 Departamento de Anatomía Patológica, Facultad de Medicina, Universidad Autónoma de Madrid (UAM), Madrid, Spain, 5 Instituto de investigaciones sanitarias del hospital La Paz (IdiPAZ), Madrid, Spain, 6 Centro de Investigación Biomédica en Red en Enfermedades Raras (CIBERER), Facultad de Medicina, Universidad Autónoma de Madrid, Madrid, Spain, 7 Instituto de Investigación Sanitaria Hospital 12 de Octubre (imas12), Madrid, Spain

☉ These authors contributed equally to this work.
¤a Current address: National Center for Drug Research and Evaluation, Istituto Superiore di Sanità, Rome, Italy.
¤b Current address: Max Delbrück Center for Molecular Medicine (MDC), Berlin, Germany.
* jorge.martin.perez@csic.es, jmartin@iib.uam.es

**Data Availability Statement:** Data have been uploaded to Proteome Xchange database (Project

## Abstract

Deregulation of Src kinases is associated with cancer. We previously showed that SrcDN conditional expression in MCF7 cells reduces tumorigenesis and causes tumor regression in mice. However, it remained unclear whether SrcDN affected breast cancer stem cell functionality or it reduced tumor mass. Here, we address this question by isolating an enriched population of Breast Cancer Stem Cells (BCSCs) from MCF7 cells with inducible expression of SrcDN. Induction of SrcDN inhibited self-renewal, and stem-cell marker expression (Nanog, Oct3-4, ALDH1, CD44). Quantitative proteomic analyses of mammospheres from MCF7-Tet-On-SrcDN cells (data are available via ProteomeXchange with identifier PXD017789, project DOI: 10.6019/PXD017789) and subsequent GSEA showed that SrcDN expression inhibited glycolysis. Indeed, induction of SrcDN inhibited expression and activity of hexokinase, pyruvate kinase and lactate dehydrogenase, resulting in diminished glucose consumption and lactate production, which restricted Warburg effect. Thus, c-Src functionality is important for breast cancer stem cell maintenance and renewal, and stem cell transcription factor expression, effects linked to glucose metabolism reduction.

## Introduction

Mammary gland stem cells are capable of self-renewal and to generate all differentiated progeny of the gland [1, 2]. Because of their long lifespan, they can accumulate mutations to become breast cancer stem cells (BCSCs). BCSCs are slow dividing and represent 1–2% of a

accession number: PXD017789, project DOI: 10.6019/PXD017789).

**Funding:** This work has been supported by grand SAF2016–75991-R (MINECO, AEI/FEDER, UE) to Jorge Martín-Pérez and ISCIII [grand PI 16/00789] to Miguel Ángel Fernández-Moreno. Víctor Mayoral-Varo was supported by the grand SAF2016–75991-R (MINECO, AEI/FEDER, UE). We acknowledge support for publication fee by the CSIC Open Access Publication Support Initiative through its Unit for Information Resources for Research (URICI).

**Competing interests:** The authors have declared that no competing of interests exist.

tumor biopsy; nevertheless, they reproduce the tumor upon transplantation in nude mice [3]. Breast tumors exhibiting basal phenotype, like the triple negative breast cancers (TNBC), are usually highly metastatic with poor prognosis, resistance to chemotherapy, and are similar to the earliest mammary gland progenitor cells, supporting the hypothesis that therapeutic failure results from inefficient BCSCs targeting [4]. In contrast, luminal A human mammary epithelial cells produce tumors in a hormone-dependent manner. As they express estrogen and progesterone receptors, but not HER2 (human epidermal growth factor 2), and they are poorly/non-metastatic, the MCF7 cell line is a cellular model representative of this type of mammary tumors [5].

The Src-family tyrosine kinases (SFKs) acts as intracellular mediators for signaling pathways from growth factors, estrogen and cytokine receptors activation, etc., mediating pathways involved in proliferation, survival, differentiation, adhesion, migration, and invasion [6–8]. SFKs is overexpressed and/or aberrantly activated in human tumors [7, 9, 10]. In breast cancer, SFKs has been associated with induction, progression, and metastasis [8, 11–16]. c-Src has been involved in glycolysis regulation and Warburg effect [17–20], which is the elevated rate of glycolysis and lactate production in aerobic conditions by tumor cells [21]. As initially shown, c-Src/v-Src phosphorylates several glycolytic enzymes, including enolase and lactate dehydrogenase (LDH) [22]. In 3T3 fibroblast, v-*src* transfection leads to increased expression of glucose uptake [23]. c-Src phosphorylates Y10-LDH, increasing its activity, which promotes invasion, anoikis resistance, and metastasis of breast cancer cells [19]. Similarly, c-Src phosphorylates hexokinase 1 and 2 (HK1, HK2) augmenting their activity, which stimulates glycolysis, cell proliferation, migration, invasion, and tumorigenesis [20]. In human glioblastoma U87 and U251 cells, EGFR activation leads to c-Src stimulation and subsequent Y59-Cdc25A phosphorylation, which dephosphorylates pyruvate kinase (PKM2), promoting its interaction with β-catenin, its transactivation and Myc transcription that induces Glut-1 (glucose transporter 1), PKM2 and LDHA expression and, consequently, the Warburg effect and tumorigenesis [18].

We previously showed in MCF7 cells that conditional Tet-On expression of SrcDN (Src-Dominant Negative) leads to inhibition of proliferation, attachment, spreading and migration in cultured cells. Inoculation of cells in nude mice generates tumors, while induction of SrcDN expression significantly reduces their tumorigenesis, and causes regression when induced in established tumors [24]. The Src-DN is a chicken paralog of c-Src with two mutations (c-Src-K295M/Y527F). The mutation K295M prevent binding of ATP to c-Src, avoiding its tyrosine kinase activity, the second point mutation Y527F at the C-terminal of the molecule, that is phosphorylated by CSK (C-terminal Src Kinase), provokes that c-Src is forced to maintain its open conformation, which implies full functionality of the SH2 and SH3 domains [6–8].

Here we addressed the question as to whether interfering SFKs functionality by SrcDN expression directly affects MCF7-BCSC renewal. Thus, here we isolate by FACS the enriched population of BCSCs (ESA$^+$-CD44$^+$-CD24$^{-/Low}$ cells, from now on CD24$^-$) and the so-called tumor-differentiated cells (ESA$^+$-CD44$^+$-CD24$^+$ cells, from now on CD24$^+$) from MCF7-Tet-On-SrcDN [24], and test their capacity of self-renewal. Our findings show that c-Src is important for mammospheres self-renewal, which is associated with an alteration in glucose metabolism.

## Materials and methods

### Reagents

Information about antibodies used in these experiments is in S2 Table in S1 File (Antibodies). BCA protein assay (Termofisher); Acrylamide/bis 40% solution, 29:1 (3.3% C), ammonium

persulfate and clarity immunoblot ECL substrate (Bio-Rad); trypan blue, doxycycline (Doxy), BSA, puromycin, and insulin (Sigma-Aldrich); EGF, and bFGF (PeproTech EC Ltd., London, UK); G418, versene, and trypsin (Invitrogen); tetracycline-free fetal bovine serum (Tet-Free-FCS, PAA Laboratories GmbH). Other chemical reagents and enzymes used were of analytical grade and purchased from Roche, GE Healthcare, or Sigma-Aldrich/Merck.

## Cells and culture

MCF7 (HTB-22) were from ATCC. Cells were mycoplasma free and authenticated by short-tandem-repeat analysis (GenePrintR 10 System from Promega, and GeneMapper v3.7 STR profile analysis software, Life Technologies) (see Supplementary Information). Profiles obtained were checked against public databases ATCC and DSMZ. MCF7-Tet-On-SrcDN cells bearing a Doxy-inducible SrcDN (avian c-Src-K295M/Y527F) were previously generated [24] and maintained in DMEM, 5% Tet-Free-FCS, 2 mM glutamine, 100 IU/mL penicillin, 100 µg/mL streptomycin, and 0.2 mg/mL G418, 0.5 µg/mL puromycin for selection.

## Isolation and culture of CD24⁻ and CD24⁺ subpopulations from MCF7 cells

The enriched subpopulations of CD24⁻ and CD24⁺ cells derived from MCF7-Tet-On-SrcDN were isolated by fluorescence-activated cell sorting (FACS), as described [3]. Briefly, $1x10^7$ cells were detached from the culture plates with versene (5 min, 37°C), and then simulta-neously labeled with antibodies to ESA-FITC, CD44-APC, CD24-PE, and with their respective isotypic immunoglobulins. Cells were washed and subjected to FACS with a FACS-Vantage cell sorter (BD, San Jose CA) equipped with an argon ion laser (emission at 488 nm) and a He-Ne laser (emission at 633 nm). Cells were gated on forward and side scatters properties and specific fluorescent signals were collected using 530 nm (FITC), 575 nm (PE) and 660 nm (APC) bandpass filters. About 1.6% of the cell population showed the CD24⁻ (ESA⁺-CD44⁺-CD24⁻) phenotype and were isolated. In parallel, the CD24⁺ (ESA⁺-CD44⁺-CD24⁺) cells were also isolated as described [3].

CD24⁻ cells were maintained in mammosphere media (1:1 DMEM/HAM'S F12, 2 mM glu-tamine, 100 IU/mL penicillin, 100 mg/mL streptomycin, 5 µg/mL Insulin, 20 ng/mL EGF, 10 ng/mL bFGF, 4 mg/ml BSA), and 0.2 mg/mL G418, 0.5 µg/mL puromycin to maintain selec-tion for c-SrcDN expression [24], and they were grown in suspension in 6-multiwell ultralow attachment plates (Falcon 351146, BD); CD24⁺ cells were cultured in DMEM, 5% Tet-Free-FCS, 2 mM glutamine, 100 IU/mL penicillin, 100 mg/mL streptomycin, and 0.2 mg/mL of G418 and 0.5 µg/mL puromycin in standard plates (Falcon, BD). Also, another protocol was employed for BCSCs enrichment to enrich for BCSCs from MCF7-Tet-On-SrcDN, single cells obtained after trypsinization of adherent cultures were plated at $1x10^3$ cells/mL and cultured as described above for CD24⁻ [25].

## Sphere Formation Efficiency (SFE)

SFE from MCF7-Tet-On-SrcDN was determined as described [26]. Briefly, single cell suspen-sions of adherent cultures were plated in 6-well ultralow attachment plates (Falcon, Corning Life Science) at $2x10^3$ cells/well and maintained in serum-free DMEM/F12 media (1:1), BSA (4 mg/mL), EGF (20 ng/mL) and bFGF (20 ng/mL), insulin (5 µg/mL), hydrocortisone (5 µg/mL) to obtain mammospheres. Fifteen days later, mammospheres were dissociated into single cells that were plated in 6-well ultralow attachment plates at $2x10^3$ cells/well in mammosphere culture media, 3 wells without Doxy (Control) and 3 wells with Doxy (2 µg/mL). During the experiments Doxy was renewed every 3 days in cultures. Each mammosphere generation was

cultured about 15 days. Mammospheres (sphere-like structures with diameter $\geq$ 50 μm) were clearly detected by optical phase contrast microscope (Nikon-Eclipse TS100, 4x magnification). For mammosphere dissociation, they were collected in a sterile tube and allow them to sediment, the media was removed with a pipette and trypsinized in Tris-Saline pipetting up and down to facilitate mammosphere dissociation, cells were then collected by centrifugation to remove trypsin, washed with mammosphere media and counted. Single cells were seeded again for mammosphere formation in 6-well ultralow attachment plates at $2x10^3$ cells/well. The experiment ended at $3^{rd}$ generation of mammosphere formation. The cells were growth in absence (Control) or presence (Doxy) of 2 μg/ml Doxy for three generations, and renewed every three days. SFE was calculated as number of spheres formed per number of seeded cells and expressed as % means ± SD. The SFE experiments were repeated 5 times, each one of them was carried out in triplicates ($p < 0.05^*$).

## Immunohistochemistry of mammospheres

Mammospheres derived from MCF7-Tet-On-SrcDN were fixed in 4% formaldehyde solution. Fixed samples were paraffin embedded, and 5 μm sections were analyzed by hematoxylin/eosin, and E-cadherin and P-cadherin staining were performed using Envision method [27].

## Immunoblotting analysis

Preparation of cell lysates from mammospheres and immunoblotting analyses were carried out as previously described [24]. Briefly, mammospheres were collected by centrifugation at 200xg 5 min at room temperature, washed twice with cool PBS and lysed at 4˚C with lysis buffer [10 mM Tris–HCl (pH 7.6), 50 mM NaCl, 30 mM sodium pyrophosphate, 5 mM EDTA, 5 mM EGTA, 0.1% SDS, 1% Triton X-100, 50 mM NaF, 0.1 mM $Na_3VO_4$, 1 mM PMSF, 1 mM benzamidine, 1 mM iodoacetamide and 1 mM phenantroline]. Cell lysates were obtained by centrifugation at 21,380xg for 30 min at 4˚C; protein concentration in the supernatant was determined by BCA protein assay (Pierce, Rockford, IL), and lysates were adjusted to equivalent concentrations with lysis buffer. Aliquots of 10–40 μg of total cell lysate were then separated on SDS–PAGE. Proteins were transferred to PVDF membranes that were blocked overnight at 4˚C with 5% non-fat milk in TTBS (TBS with 0.05% Tween-20). Incubation with primary specific antibodies was carried out overnight at 4˚C, and horseradish peroxidase-conjugated secondary antibodies in blocking solution for 1 h at room temperature. Immunoreactive bands were visualized by ECL kit.

## Quantitative proteomic analysis

Briefly, MCF7-Tet-On-SrcDN derived mammospheres at $3^{rd}$ generation from two independent experiments cultured either in the absence (Control) or presence of Doxy (2 μg/mL) were collected by centrifugation, washed with PBS, and lysed in 9 M urea, 2 M thiourea, 5% CHAPS, and 2 mM TCEP. The samples were precipitated by methanol/chloroform and quantified by Pierce 660nm Protein Assay. Then 30 μg of protein from each sample were trypsinized, and the resulting peptides were labeled using iTRAQ 4-plex reagent (SCIEX, Foster City, CA, USA) according to the manufacturer's instructions and named as follows: Control-1, 114-tag; Doxy-1, 115-tag; Control-2, 116-tag; Doxy-2, 117-tag. Samples were pooled, evaporated to dryness and frozen. The pool was separated into 5 fractions in a reverse phase C18 chromatography at basic pH and cleaned with a StageTip-C18. The 5 fractions were quantified by fluorimetry, and 1 μg of each fraction was separated by C-18 reverse phase chromatography during 150 min. The eluted peptides passed then into 5600 Triple-TOF MS (SCIEX, Foster City, CA). The raw data of the combined 5 fractions was exported to the search-engine Mascot

Server v2.5.1 (Matrix Science, London, UK) confronted to the *Homo sapiens* UniProt database. Peptide mass tolerance was set to ± 25 ppm for precursors and 0.05 Da for fragment masses. Frequency distribution histograms of protein ratios were obtained into Excel 2010. Log2 peptide ratios followed a normal distribution that was fitted using least squares regression. Mean and standard deviation values derived from the Gaussian fit were used to calculate p-values and quantification of False Discovery Rates (FDR). The confidence interval for protein identification was set to ≥ 95% (p<0.05) and only peptides with an individual ion score above the 1% False Discovery Rates (FDR) threshold were considered correctly identified. Only proteins having at least two quantitated peptides were considered in the quantitation. A 5% quantitation FDR threshold was estimated to consider the significant differentially expressed proteins.

The mass spectrometry proteomics data have been deposited to the ProteomeXchange Consortium via the PRIDE [28] partner repository (Project accession number: PXD017789, project DOI:10.6019/PXD017789).

## Enzymatic activity assays

Mammospheres at 3rd generation derived from MCF7-TET-ON-SrcDN cells were seeded into a six-well plate and cultured with fresh medium -/+ Doxy (2 μg/mL) for 12–24 h. Cells were harvested by centrifugation, washed twice with PBS and preincubated for 10 min in PBS. Then, cells were collected by centrifugation, suspended in fresh culture medium and incubated for another 4 h. Cells were harvested by centrifugation at room temperature at 0 and 4 h incubation and treated with 200 μL lysis buffer (Tris-HCl (pH 7.5), 1 mM EDTA, 0.5 mM dithiothreitol, 0.2% Triton X-100, 0.2 mg/mL deoxycholate, 0.2 mM phenylmethylsulfonyl fluoride, and 2.5 μg/mL leupeptin), and homogenized at 4°C by pipetting up and down. The homogenate was centrifuged at 21,380x*g* for 15 min at 4°C and the supernatant was used for enzymatic activity measurements. Hexokinase (HK) activity was determined spectrophotometrically at 340 nm by following the G6PDH dependent conversion of NADP to NADPH. The assay mixture contained 100 mM Tris-HCl (pH 8.0), 0.5 mM EDTA, 10 mM $MgCl_2$, 5 mM MgATP, 2 mM glucose, 0.5 mM $NADP^+$ and 10 U/mL glucose 6-phosphate dehydrogenase (G6PDH) in a final volume of 1 mL. HK activities were calculated according to the slope of the resulting curves in the log phase. The initial velocity of lactate dehydrogenase (LDH) was determined by measuring spectrophotometrically the NADH consumption at 340 nm in a 1 mL reaction mixture (50 mM HEPES pH 7.2, 100 mM KCl, 5 mM $MgCl_2$, 0.15 mM NADH, and 2 mM pyruvate). The slopes of the initial linear decrease were used to calculate initial velocities. Pyruvate kinase (PK) activity was measured by following the production of pyruvate. The reaction mixture consisted of 50 mM HEPES (pH 7.2), 100 mM KCl, 5 mM $MgCl_2$, 0.15 mM NADH, 1 mM MgADP, 2 mM phosphoenolpyruvate, and 1 U/mL of LDH as the auxiliary enzyme in a final volume of 1 mL. PK activity was determined spectrophotometrically as above by recording the change of 340 nm absorbance produced by NADH oxidation. One unit of enzyme activity is defined as the amount of enzyme catalyzing the conversion of 1 μmol of substrate/min at 25°C. Enzymatic activities of different samples were normalized against the corresponding protein concentrations in the lysates.

## Lactate production and glucose consumption assays

Cells were treated as above. At 0 and 4 h incubation, cells were harvested by centrifugation at room temperature, and the culture medium was removed and stored at -70°C for lactate production and glucose consumption measurements. Cell pellets were homogenized in ice-cold 8% (v/v) perchloric acid in 40% ethanol. After centrifugation at 21,380x*g* for 15 min at 4°C, the supernatant was neutralized to pH 7.0 with a 2 M KOH and 0.5 M triethanolamine solution,

the mixture was incubated on ice for 15 min, and precipitated $KClO_4$ was removed by centrifugation. Glucose and lactate content in culture medium and levels of intracellular lactate were assessed by using standard enzymatic techniques [29]. Briefly, lactate was determined in a reaction containing 0.5 M Glycine, 0.2 M hydrazine, 3.4 mM EDTA, 1.5 mM $NAD^+$, and 10 U/mL of LDH, pH 9.5. The reaction mixture for glucose measurement consisted of 0.1 M triethanolamine, 5 mM $MgCl_2$, 0.5 mM $NADP^+$, 1 mM ATP, 1 U/mL hexokinase and 3 U/mL G6PD, pH 7.4. The function of the glycolytic pathway was evaluated by the amount of lactate produced per hour after 4 h incubation. Glucose consumption rate (per hour) was calculated considering the difference between the amount in the medium at 0 h and that recovered in the medium after 4 h incubation. Glucose and lactate content were normalized to the protein concentration of cells cultured under similar conditions and obtained as described above for enzymatic activity.

## Oxygen consumption and determination of ATP concentration

Mammospheres at $3^{rd}$ generation cultured in the absence or presence of 2 μg/mL Doxy were dissociated to single-cell suspension as for SFE, resuspended in mammospheres media with or without pyruvate and without BSA to $10^6$ cells/mL and the oxygen consumption of $2 \times 10^6$ cells was measured by high-resolution respirometry with the OROBOROS oxygraph-2k in a standard configuration, at 37°C and a 750-rpm stirrer speed as described [30]. The protocol includes the determination in a sequential manner of the aerobic metabolic activity under routine culture conditions with the physiological substrates in culture medium (Cr), the oligomycin-inhibited leak rate of respiration after inhibition of ATP synthase with 2 μg/mL oligomycin (CrO) and the maximum respiratory capacity of uncoupled mitochondria in non-permeabilized cells (CrU), obtained by sequential addition of 0.5 μM boluses of trifluoro-carbonylcyanide phenylhydrazone (FCCP). The protonophore FCCP dissipates the proton inner membrane gradient making ETC to function at its maximal rate since oxygen and protons are not limiting for the ETC Complex IV catalysis. All values, including physiological conditions and the maximal respiratory capacity, are derived by subtracting non-mitochondrial respiration from the FCCP rate.

Measurements of total ATP concentration were made in mammospheres that were maintained for at least 15 days -/+ Doxy (2 μg/mL) at 0 and 4 h, as previously described [31], following instructions of the ATP Bioluminescence Assay Kit CLS II (Roche, Ref. 11699695001).

## Statistical analyses

Mean values, standard deviation and statistical significance between data from two different experimental conditions were determined by two-tail Student *t*-test. In all figures, data represent the Mean from at least three separate biological repeats done in at least triplicates each ± SD (standard deviation), $^*p < 0.05$, $^{**}p < 0.01$, and $^{***}p < 0.001$.

## Results

### Isolation and characterization of an enriched population of BCSCs from MCF7 cells

To determine the role of SrcDN in the self-renewal of the enriched subpopulation of BCSCs, we isolated $CD24^-$ ($ESA^+$-$CD44^+$-$CD24^-$) and $CD24^+$ ($ESA^+$-$CD44^+$-$CD24^+$) cell populations derived from MCF7-Tet-On-SrcDN by FACS. Cells were first sorted as $ESA^+$ and then for being $CD44^+$-$CD24^-$ (1.6%), while the $CD44^+$-$CD24^+$ represented 98.4% of total cell population (Fig 1A, isotype controls shown in S1 Fig in S1 File). $CD24^-$ cells formed mammospheres

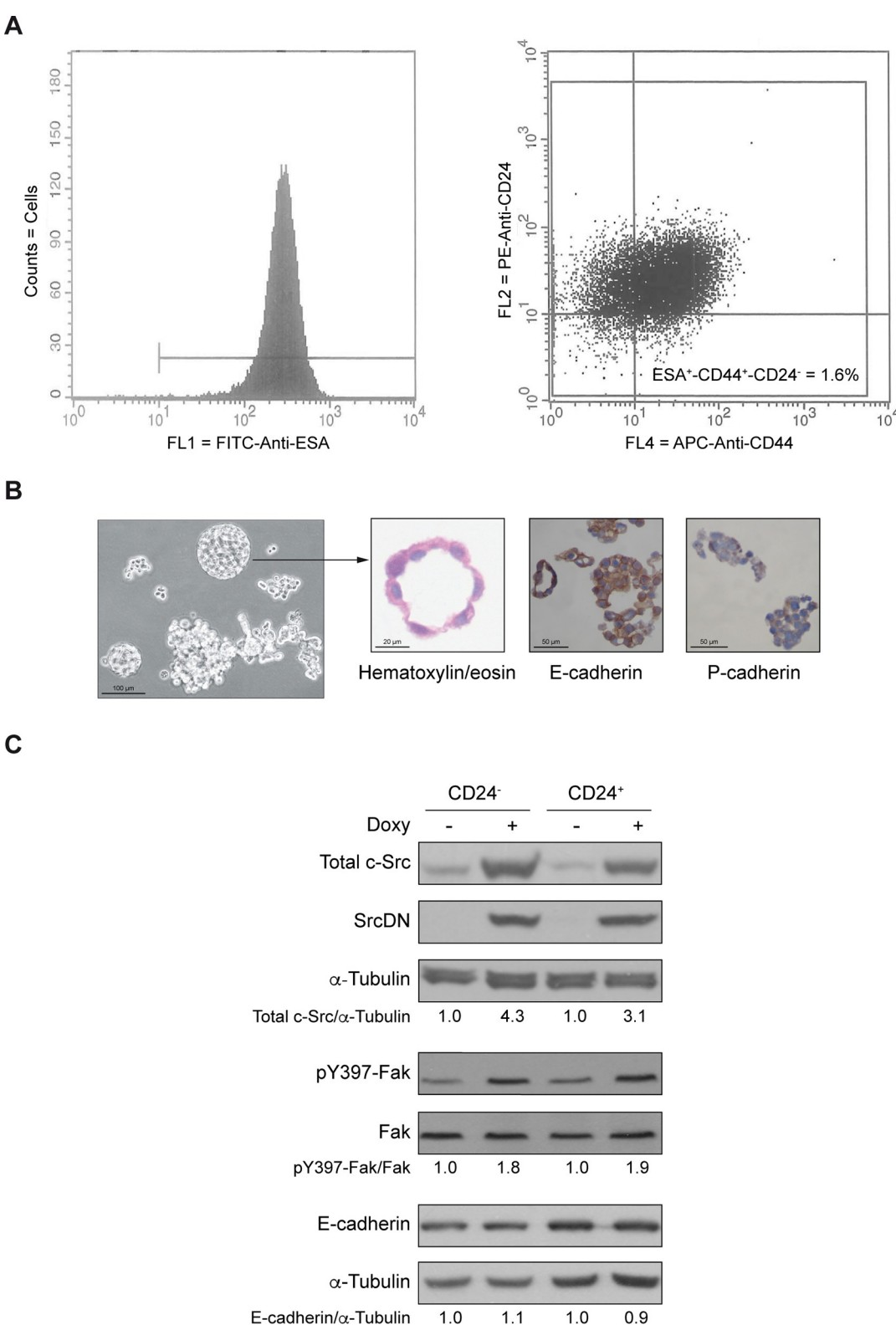

**Fig 1. Cell sorting of CD24⁻ and CD24⁺ subpopulations of MCF7 cells.** (A) Histograms of ESA-FITC positive cells that were later sorted for CD44-APC positive and CD24-PE negative (1.6%). (B) Microphotograph under light microscope of mammospheres from CD24⁻ cells in culture. Representative images of spheres stained with hematoxylin/eosin, E-cadherin or P-cadherin. Scales bars are included. (C) Immunoblot of total c-Src, SrcDN, p-Y397-Fak/Fak, and E-cadherin, using α-Tubulin

expression as loading control from CD24- cells forming mammospheres and CD24+ cells in absence (Control) or presence of Doxy (2 μg/mL) for induction of SrcDN during three passages. The ratios of the proteins Total c-Src, SrcDN, and E-cadherin with the loading control α-Tubulin, or pY397-Fak/Fak were calculated and referred to -Doxy (Control) considered as 1. These are representative results from 3 independent experiments.

(Fig 1B) that after staining with hematoxylin/eosin showed a monolayer cell-structure with their nuclei oriented towards the lumen. Furthermore, mammospheres expressed E-cadherin, but not P-cadherin (Fig 1B). These results suggested that both CD24- (mammospheres) and CD24+ cells maintained the epithelial phenotype of the MCF7 total population.

To determine the functionality of SrcDN, protein extracts from mammospheres from CD24- and from CD24+ cell subpopulations were blotted for pY397-Fak, showing an increased phosphorylation upon SrcDN induction. This is consistent with previous results showing that SH2 domain from SrcDN binds pY397-Fak protecting from dephosphorylation [24, 32, 33], and with the reduction of pY925-Fak, p-Y-p130CAS, pY118-Paxillin phosphorylation by c-Src in MCF7 cells [24]. Expression of E-cadherin remained unaltered in CD24- and CD24+ upon SrcDN induction (Fig 1C). SrcDN expression was tested by detecting total c-Src and avian SrcDN (Fig 1C).

## c-Src functionality is necessary for mammospheres self-renewal

To characterize the role of c-Src in the enriched subpopulation of BCSC self-renewal, we analyzed whether induction of SrcDN expression altered formation of mammospheres from MCF7-Tet-On-SrcDN. Self-renewal capacity was confirmed by three successive generations of mammospheres, where a progressive enrichment of mammosphere-forming cells (CD24-) was observed (S2 Fig in S1 File). Then, we determined Sphere Formation Efficiency (SFE) upon induction of SrcDN expression from the first generation and found that it significantly reduced SFE as compared to control (-Doxy) at the 3rd generation (Fig 2A). This effect was

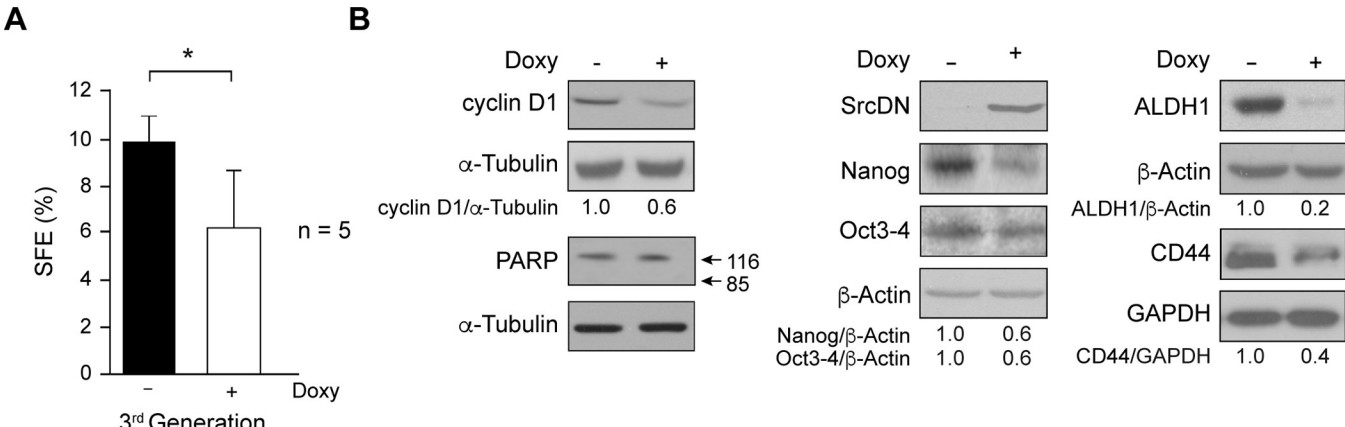

**Fig 2. Self-renewal of mammospheres derived from MCF7.** (A) The Sphere Formation Efficiency (SFE) was determined in MCF7-Tet-On-SrcDN. Thus, single cell suspensions from adherent cells were plated in 6-well ultralow attachment plates at $2x10^3$ cells/well and cultured in mammosphere medium. Fifteen days later, mammospheres were dissociated into single cells that were divided into two groups: Control (-Doxy) and Doxy treated (2 μg/mL). Doxy-treatment was maintained for three generations, and renewed every 3 days. SFE at 3rd generation was calculated as number of spheres formed per number of seeded cells and expressed as % means ± SD. The SFE experiments were repeated 5 times, each one of them was carried out in triplicates. ($p < 0.05^*$). (B) Total cell extracts from the 3rd generation of mammospheres treated as above (panel A) were used to determine by immunoblotting expression of cyclin D1, PARP, with α-Tubulin, as a loading control, and SrcDN, Nanog, Oct3/4, ALDH1, and CD44, employing GAPDH or β-Actin, as a loading control. The net quantification of the gel bands after subtracting the background was carried out with ImageJ program and expressed in arbitrary units. These are representative results from 3 independent experiments. The ratios of the proteins SrcDN, Nanog, Oct3/4, ALDH1, and CD44 and their loading controls GAPDH or ß-Actin, or pY10-LDHA/LDHA were calculated and referred to -Doxy (Control) considered as 1.

clearly evident from the 1st generation (S2 Fig in S1 File), although no morphological alterations were observed upon SrcDN induction (data not shown). Consistently, cyclin D1 expression at the 3rd generation was also reduced. However, SrcDN expression did not appear to cause apoptosis, as PARP degradation was undetectable. To determine the nature of cells that form mammospheres, we have dissociated mammospheres at the 3rd generation, labelled cells with anti-CD24-PE and CD44-APC monoclonal antibodies, as well as with isotypic immunoglobulins for control, and analyzed them by flow cytometry. S3 Fig in S1 File showed that in control mammospheres most of cells are CD44+-CD24-. In contrast, in Doxy-treated mammospheres a large number of cells were CD44+-CD24+. In addition, the analysis of the expression level of these markers (MFC, mean fluorescence channel) indicates that, along with an increase of expression of CD24, treated mammospheres show a reduced expression of CD44. Moreover, expression of pluripotency-related transcription factors involved in self-renewal Nanog, Oct3/4, and stem cell markers ALDH1, and CD44 [34–36] were also reduced (Fig 2B). Together, these results suggested that c-Src functionality is relevant for MCF7-mammosphere renewal.

## Proteomic analyses of mammospheres

Since c-Src functionality is important for SFE of MCF7-mammospheres, we analyzed whether SrcDN induction quantitatively modified protein expression. Proteomic analyses identified 2,759 proteins with at least two peptides, full data available at Proteome Xchange database (project accession number: PX017789, project DOI: 10.6019/PX017789). From those, only 101 proteins were differentially expressed with an FDR < 5% (S1 Table in S1 File). In the Fig 3 only proteins having at least two quantitated peptides and <5% quantitation FDR with the same tendency in both experiments were included. Thus, induction of SrcDN by Doxy (2 μg/mL) diminished the levels of fifteen proteins, and increased the levels of four of them (Fig 3, and S4 Fig in S1 File). The most increased protein in Doxy/Control is c-Src, which most likely should correspond to SrcDN induced by Doxy-treated mammospheres, as the identity of *Homo Sapiens* and *Gallus-Gallus* c-Src at the amino acid sequences is 94%. Results obtained were submitted to GSEA and found that glycolysis was significantly reduced in MCF7-mammospheres upon SrcDN induction (S5 Fig in S1 File), which was consistent with data from Fig 3.

| DIFFERENTIAL PROTEINS FOR 5% FDR AT QUANTITATION LEVEL | | Doxy-18/Control-18 | | | Doxy-29/Control-29 | | |
|---|---|---|---|---|---|---|---|
| Protein_AC | Description | Ratio | Log2 | FDR | Ratio | Log2 | FDR |
| O14737 | Programmed cell death protein 5 OS=Homo sapiens GN=PDCD5 PE=1 SV=3 | #N/A | #N/A | #N/A | #N/A | #N/A | #N/A |
| P00338 | L-lactate dehydrogenase A chain OS=Homo sapiens GN=LDHA PE=1 SV=2 | 0.589 | -0.764 | 2.36% | 0.567 | -0.818 | 0.04% |
| P00558 | Phosphoglycerate kinase 1 OS=Homo sapiens GN=PGK1 PE=1 SV=3 | 0.583 | -0.777 | 2.05% | 0.569 | -0.813 | 0.04% |
| P05161 | Ubiquitin-like protein ISG15 OS=Homo sapiens GN=ISG15 PE=1 SV=5 | 0.540 | -0.889 | 0.54% | 0.337 | -1.568 | 0.00% |
| P09104 | Gamma-enolase OS=Homo sapiens GN=ENO2 PE=1 SV=3 | 0.541 | -0.885 | 0.56% | 0.642 | -0.639 | 0.94% |
| P09972 | Fructose-bisphosphate aldolase C OS=Homo sapiens GN=ALDOC PE=1 SV=2 | 0.569 | -0.812 | 1.37% | 0.612 | -0.708 | 0.31% |
| P13674 | Prolyl 4-hydroxylase subunit alpha-1 OS=Homo sapiens GN=P4HA1 PE=1 SV=2 | 0.572 | -0.807 | 1.41% | 0.653 | -0.616 | 1.45% |
| P18085 | ADP-ribosylation factor 4 OS=Homo sapiens GN=ARF4 PE=1 SV=3 | 0.494 | -1.017 | 0.09% | 0.684 | -0.549 | 4.15% |
| P19525 | Interferon-induced, double-stranded RNA-activated protein kinase OS=Homo sapiens GN=EIF2AK2 PE=1 SV=2 | 0.571 | -0.807 | 1.43% | 0.594 | -0.751 | 0.13% |
| P52789 | Hexokinase-2 OS=Homo sapiens GN=HK2 PE=1 SV=2 | 0.546 | -0.874 | 0.67% | 0.428 | -1.226 | 0.00% |
| P61006 | Ras-related protein Rab-8A OS=Homo sapiens GN=RAB8A PE=1 SV=1 | 0.585 | -0.773 | 2.14% | 0.606 | -0.722 | 0.24% |
| P84077 | ADP-ribosylation factor 1 OS=Homo sapiens GN=ARF1 PE=1 SV=2 | 0.363 | -1.464 | 0.00% | 0.559 | -0.838 | 0.03% |
| P84085 | ADP-ribosylation factor 5 OS=Homo sapiens GN=ARF5 PE=1 SV=2 | 0.365 | -1.452 | 0.00% | 0.497 | -1.009 | 0.00% |
| Q8N5K1 | CDGSH iron-sulfur domain-containing protein 2 OS=Homo sapiens GN=CISD2 PE=1 SV=1 | 0.567 | -0.818 | 1.36% | 0.636 | -0.652 | 0.75% |
| Q9H299 | SH3 domain-binding glutamic acid-rich-like protein 3 OS=Homo sapiens GN=SH3BGRL3 PE=1 SV=1 | 0.550 | -0.862 | 0.74% | 0.454 | -1.139 | 0.00% |
| *P12931* | *Proto-oncogene tyrosine-protein kinase Src OS=Homo sapiens GN=SRC PE=1 SV=3* | *3.079* | *1.622* | *0.00%* | *2.738* | *1.453* | *0.00%* |
| *P42695* | *Condensin-2 complex subunit D3 OS=Homo sapiens GN=NCAPD3 PE=1 SV=2* | *2.595* | *1.376* | *0.00%* | *2.776* | *1.473* | *0.00%* |
| *Q3ZCQ8* | *Mitochondrial import inner membrane translocase subunit TIM50 OS=Homo sapiens GN=TIMM50 PE=1 SV=2* | *1.769* | *0.823* | *4.91%* | *1.673* | *0.743* | *1.65%* |
| *Q93070* | *Ecto-ADP-ribosyltransferase 4 OS=Homo sapiens GN=ART4 PE=2 SV=2  CD297* | *2.120* | *1.084* | *0.18%* | *1.746* | *0.804* | *0.59%* |

| Down-expressed | | Significant (FDR<1%) | |
|---|---|---|---|
| Up-expressed | | Significant (1%<FDR>5%) | |

**Fig 3. Quantitative proteomic analysis of mammospheres derived from MCF7-Tet-On-SrcDN.** The analysis was carried out in two independent experiments from the total cell extract of the 3rd generation mammospheres either untreated (Control) or Doxy-treated. Only proteins having at least two quantitated peptides and <5% quantitation FDR in both experiments are included in the figure.

## Role of c-Src in the regulation of glycolysis in MCF7-mammospheres

As glycolytic pathway was inhibited in mammospheres upon SrcDN induction, we investigated whether glucose consumption was altered, and found that it was significantly reduced in mammospheres expressing SrcDN (Fig 4A), which parallels a reduction in Glut-1 levels (Fig 4D), the major glucose transporter in tumor cells [37]. In cancer cells, Src enhances HK enzymatic activity, a rate-limiting enzyme of glycolysis, by direct phosphorylation [20]. Since SrcDN is a kinase-dead mutant and competes with endogenous Src kinases [38], HK enzymatic activity was significantly inhibited in Doxy-treated mammospheres (Fig 4C). In addition, HK2 levels were also diminished (Fig 4D), supporting proteomic results (Fig 3). However, we were unable to detect HK2 tyrosine phosphorylation. Similarly, PKM enzymatic activity was also inhibited upon SrcDN expression (Fig 4C). In head and neck and in breast cancer cells, Src phosphorylates and enhances LDHA [19]. We tested for SrcDN effects on LDHA enzymatic activity, expression, and phosphorylation. In Doxy-treated mammospheres, LDH activity was significantly reduced (Fig 4C), as it was pY10-LDHA/LDHA ratio (Fig 4D). It is well established that c-Myc and HIF-1 promote the expression of enzymes implicated in glycolysis regulation, including HK2 and LDHA [39, 40, 41]. Thus, we analyzed c-Myc and HIF-1 expression level in Control (Doxy-untreated) and Doxy-treated (2 μg/mL) 3rd mammosphere generations by SYBR Green-q-RT-PCR and by immunoblotting (see Supplementary Material). SrcDN induction did not modify c-Myc expression both at RNA and protein level while HIF-1 RNA expression was significantly reduced. In addition, we were not able to detect HIF-1 protein expression by immunoblotting as the protein is extremely labile (S6 Fig in S1 File). The decrease in glucose consumption was paralleled by a nearly stoichiometric reduction of lactate production (lactate/glucose ratio of $1.45 \pm 0.12$ and $1.58 \pm 0.09$ for wild-type and SrcDN-mammospheres, respectively). Thus, the production of lactate from glutaminolysis is very limited. Interestingly, although lactate production diminished, intracellular accumulation of this metabolite was increased (Fig 4B), a result consistent with the reduced expression of MCT-1 (SLC16A1, proton coupled monocarboxylate transporter 1) in Doxy-treated mammospheres (Fig 4D), the major lactate transporter in tumor cells [42].

## Effects of SrcDN on oxygen consumption and ATP production

Results showed that SrcDN expression inhibited glycolysis and altered the lactate levels in mammospheres. We next determined the effects of SrcDN in mitochondrial respiration capacity, either in culture without pyruvate, and in normal culture media, which containing saturating concentrations of pyruvate to the uncoupled mitochondria to glycolysis, pushing oxygen consumption to its highest. Basal oxygen consumption of cells dissociated from mammospheres in media containing pyruvate was unmodified by SrcDN expression, while the maximum was significantly inhibited (Fig 5A). Similarly, oxygen consumption carried out in media without pyruvate was also unaltered in basal conditions, but the maximum was highly reduced in mammospheres expressing SrcDN (Fig 5A, and S7 Fig in S1 File: OCR profiles). Furthermore, analyses of ATP production did not differ between cells dissociated from mammospheres expressing SrcDN and control cells (Fig 5B). Neither alteration in ROS level, nor modification of catalase or MnSOD expression was observed upon SrcDN induction in mammospheres, as compared to controls (Doxy-untreated) (S8 Fig in S1 File).

## Discussion

Cancer stem cells, which represent only a small subpopulation in a tumor, possess the ability of self-renewal and to form new tumors in nude mice. In breast cancer, BCSCs are defined by expression of ESA, CD44 and low CD24 expression (ESA+/CD44+/CD24-), and high

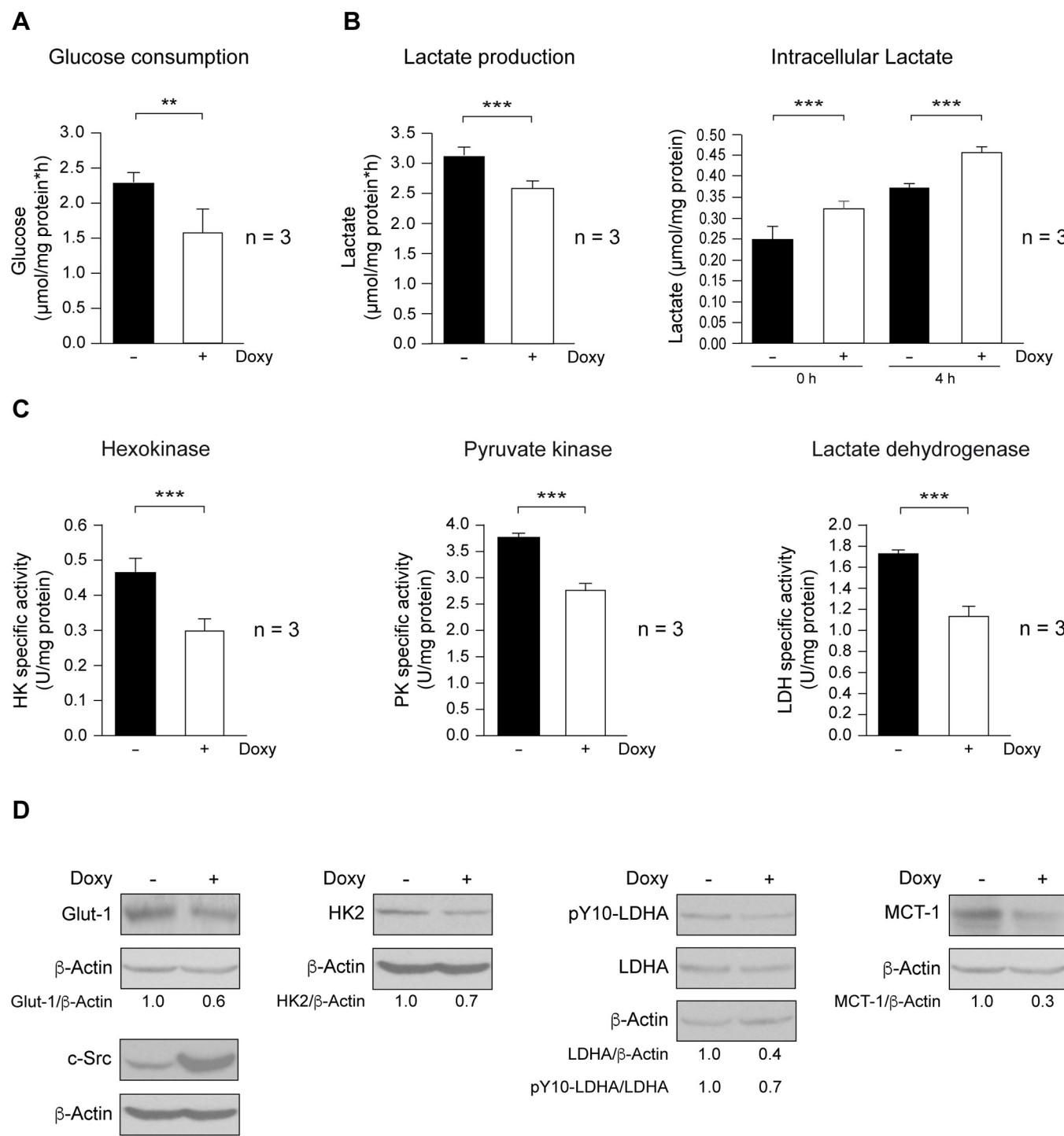

**Fig 4. Expression of SrcDN reduced glycolysis in MCF7 mammosphere-derived cells.** Control and Doxy-treated (2 µg/mL) mammospheres from the 3[rd] generation were used to study glycolysis. (A) Determination of glucose consumption. (B) Lactate production, intracellular lactate. (C) Enzymatic activity of hexokinase (HK), pyruvate kinase (PK), lactate dehydrogenase (LDH). (***p<0.001). (D) Expression of Glut-1, total c-Src, HK2, pY10-LDHA, LDHA, and MCT-1, was determined by immunoblotting, employing ß-Actin as a loading control. These are representative results from 3 independent experiments. The ratios refer to -Doxy considered as 1.

**A**

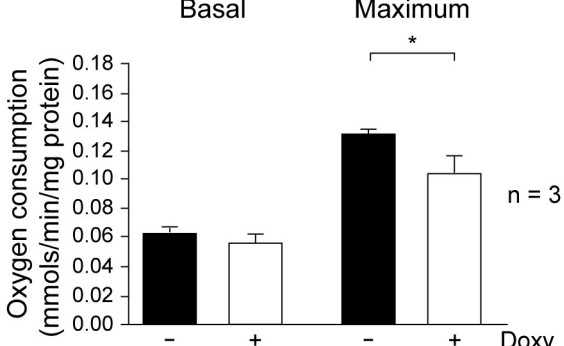
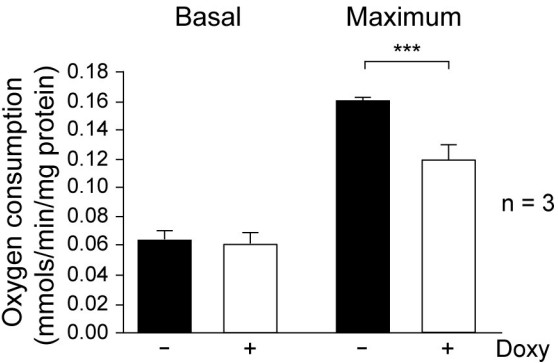

**B**

**Fig 5. Effects of SrcDN on oxygen consumption, and ATP production in MCF7 mammosphere-derived cells.** (A) Oxygen consumption in normal media containing pyruvate or in media without pyruvate and (B) ATP production, they were determined in cells obtained from Control (Doxy-untreated) and Doxy-treated (2 μg/mL) 3$^{rd}$ generation mammospheres.

tumorigenic capabilities [3]. In this context, the low percentage of BCSCs (CD24$^-$ cells) that we found here, correlates with previous observations [3] and confirms that the isolation of the CD24$^-$ cells was properly done.

In this study we analyzed the effects of the induction of a dominant-negative mutant of the protooncogene c-Src, the SrcDN. We found that disrupting c-Src pathways by induction of SrcDN expression reduced the self-renewal capabilities (SFE) of the enriched population of BCSCs derived from MCF7. Interestingly, cytometric analyses of SrcDN expressing mammosphere cells showed a reduction of CD44 and increased CD24 expression as compared to control mammosphere cells (non-expressing SrcDN). In addition, SrcDN induction did not induce apoptosis as PARP degradation did not occur, instead cyclin D1 expression was reduced, while Myc expression was unaltered. In this respect, we previously observed that SrcDN expression and selective c-Src tyrosine-kinase inhibitors diminish cyclin D1 expression and cell cycle progression at G1-phase in breast cancer cells [24, 43, 44]. Moreover, these mammosphere cells also showed expression of Oct3/4, Nanog pluripotency-associated transcription factors, ALDH1, and CD44 stem-cell markers [34–36], which were reduced upon SrcDN

induction. These pluripotency-associated transcription factors are required for embryonic stem cell self-renewal [34], and they are also involved in tumorigenesis and metastasis [45]. Specifically, MDA-MB-231 cells exhibit high expression of Oct3/4, and low levels of Nanog compared to MCF7 and T47D [46]. Collectively, our results suggest that c-Src functionality is essential for the self-renewal ability of BCSCs derived from MCF7 to grow in vitro as mammospheres.

MCF7 cells have been used as a model for Luminal A type of hormonal-dependent breast tumors, they have epithelial characteristics, they express E-cadherin, and are low or non-metastatic [5]. Here, we showed that mammosphere-forming cells derived from MCF7, which represented about 1.6% of total cell population, maintained their epithelial phenotype as they showed E-cadherin, but not P-cadherin staining.

Quantitative proteomic analyses of untreated and Doxy-treated mammospheres identified up to 2.759 proteins; of these proteins, 101 were differentially expressed (FDR < 5%), but only 19 of them showed same tendency in two independent experiments. Among those proteins, 15 showed a reduced expression upon SrcDN induction. The gene set enrichment analysis (GSEA) of the reduced proteins revealed an inhibition of the glycolytic pathway in these tumor-initiating cells. Thus, SrcDN restricts the Warburg effect in MCF7-mammospheres, as it reduced expression of Glut-1, glucose consumption, and expression and activity of HK2, PKM, LDHA, and lactate production. Since HIF-1 and Myc are known transcription factors involved in the control of HK2 and LDHA expression [39–41], we tested whether they were modulated by induction of SrcDN. While SrcDN inhibited expression of mRNA expression of HIF-1, Myc expression both at mRNA and protein levels was not altered. In human colon adenocarcinoma LS174T and in murine melanoma B16-F10 cell lines Warburg effect requires expression of both LDH A and B [47]. Interestingly, while LDH B is expressed in TNBC cells, it is undetectable in luminal A cells, such as MCF7 [48]. Since MCT-1 expression was also inhibited upon SrcDN induction in MCF7-mammospheres, lactate accumulated intracellularly. It has been recently shown that SrcDN reduced glucose catabolism required for proliferation, migration, invasion, and tumorigenesis in cancer cells [49]. c-Src phosphorylates HK1 and HK2, rate-limiting enzymes of glycolysis, and enhances their catalytic activity [20]. Moreover, c-Src also phosphorylates LDHA at Y10, increasing its activity, which correlates with metastasis progression of clinical tumor samples of breast cancer [19], while as shown here, the dominant-negative variant of c-Src, the SrcDN counteracted these effects. In contrast, the basal oxygen consumption, ATP production, ROS, catalase, and MnSOD levels were unmodified by SrcDN expression in mammospheres, indicating that under basal conditions, the mitochondrial functionality was unaltered by SrcDN expression.

These data strongly suggest that c-Src functionality is important for the self-renewal capacity in MCF7 breast cancer stem cells, linked to glucose metabolism.

## Conclusions

The findings presented in our study support the role of c-Src in the self-renewal of the enriched subpopulation of MCF7 breast cancer stem cells (BCSCs). Furthermore, results showed that c-Src functionality is a requisite for BCSCs. Src regulates Warburg effect in BCSCs, which in turn somehow controls stem-cell factors mediating stem-cell renewal. Elucidating the pathways that regulate BCSC functionality is fundamental to achieve their specific targeting.

## Supporting information

**S1 File.**
(PDF)

## Acknowledgments

Authors are grateful to L. del Peso for the GSEA of proteomics, S. Alcalá, B. Sainz, S. Guerra, C. Gamallo, P. Lastres, M.C. Mena, E. Martín-Forero, and J. Pérez for their contributions and support, L. Boscá, and J. Gonzalez-Castaño for their comments, suggestions, and reagents. Jorge Martín-Pérez and David Hardisson are members of the GEICAM (Grupo Español de Investigación en Cáncer de Mama).

## Author Contributions

**Conceptualization:** Jorge Martín-Pérez.

**Data curation:** Jorge Martín-Pérez.

**Formal analysis:** Jorge Martín-Pérez.

**Funding acquisition:** Miguel Ángel Fernández-Moreno, Jorge Martín-Pérez.

**Investigation:** Víctor Mayoral-Varo, Annarica Calcabrini, María Pilar Sánchez-Bailón, Óscar H. Martínez-Costa, Cristina González-Páramos, Sergio Ciordia, David Hardisson, Juan J. Aragón, Miguel Ángel Fernández-Moreno, Jorge Martín-Pérez.

**Methodology:** Víctor Mayoral-Varo, Annarica Calcabrini, María Pilar Sánchez-Bailón, Óscar H. Martínez-Costa, Cristina González-Páramos, Sergio Ciordia, David Hardisson, Juan J. Aragón, Miguel Ángel Fernández-Moreno, Jorge Martín-Pérez.

**Project administration:** Jorge Martín-Pérez.

**Resources:** Miguel Ángel Fernández-Moreno, Jorge Martín-Pérez.

**Supervision:** Jorge Martín-Pérez.

**Validation:** Jorge Martín-Pérez.

**Visualization:** Jorge Martín-Pérez.

**Writing – original draft:** Jorge Martín-Pérez.

**Writing – review & editing:** Víctor Mayoral-Varo, Annarica Calcabrini, María Pilar Sánchez-Bailón, Óscar H. Martínez-Costa, Sergio Ciordia, Juan J. Aragón, Miguel Ángel Fernández-Moreno, Jorge Martín-Pérez.

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
