## [Decision Letter · Decision Letter 0]

1 Apr 2020

PONE-D-20-06055

c-Src functionality controls self-renewal and glucose metabolism in MCF7 breast cancer stem cells

PLOS ONE

Dear Dr. Martín-Pérez,

Thank you for submitting your manuscript to PLOS ONE. After careful consideration, we feel that it has merit but does not fully meet PLOS ONE’s publication criteria as it currently stands. Therefore, we invite you to submit a revised version of the manuscript that addresses the points raised during the review process.

As you could see from attached comment the Reviewers several points that need to be addressed basically concerning the Figures 2 and 3.

We would appreciate receiving your revised manuscript by May 16 2020 11:59PM. To enhance the reproducibility of your results, we recommend that if applicable you deposit your laboratory protocols in protocols.io, where a protocol can be assigned its own identifier (DOI) such that it can be cited independently in the future. For instructions see: http://journals.plos.org/plosone/s/submission-guidelines#loc-laboratory-protocols

We look forward to receiving your revised manuscript.

Kind regards,

Antimo Migliaccio, M.D.

Academic Editor

PLOS ONE

Journal Requirements:

3. PLOS ONE now requires that authors provide the original uncropped and unadjusted images underlying all blot or gel results reported in a submission’s figures or Supporting Information files. This policy and the journal’s other r equirements for blot/gel reporting and figure preparation are described in detail at https://journals.plos.org/plosone/s/figures#loc-blot-and-gel-reporting-requirements and https://journals.plos.org/plosone/s/figures#loc-preparing-figures-from-image-files. When you submit your revised manuscript, please ensure that your figures adhere fully to these guidelines and provide the original underlying images for all blot or gel data reported in your submission. See the following link for instructions on providing the original image data: https://journals.plos.org/plosone/s/figures#loc-original-images-for-blots-and-gels.

4. Please include your tables as part of your main manuscript and remove the individual files. Please note that supplementary tables (should remain/ be uploaded) as separate "supporting information" files

Additional Editor Comments (if provided):

Reviewers' comments:

Reviewer's Responses to Questions

**Comments to the Author**

1. Is the manuscript technically sound, and do the data support the conclusions?

Reviewer #1: Yes

Reviewer #2: No

2. Has the statistical analysis been performed appropriately and rigorously? 

Reviewer #1: N/A

Reviewer #2: I Don't Know

3. Have the authors made all data underlying the findings in their manuscript fully available?

Reviewer #1: Yes

Reviewer #2: No

4. Is the manuscript presented in an intelligible fashion and written in standard English?

Reviewer #1: Yes

Reviewer #2: Yes

5. Review Comments to the Author

Reviewer #1: The authors have previously constructed a line of MCF7 human breast cancer cells (MCF7-Tet-On-SrcDN cells) in which expression of a dominant-negative form of the avian c-Src protein tyrosine kinase (SrcDN) can be induced by doxycycline (DOX) treatment, and shown that SrcDN expression inhibited cell attachment, spreading, and migration and also reduced tumorigenesis. Here, they have extended their studies of the phenotypic consequences of inducibly expressed SrcDN, and examined the effect SrcDN expression on the stem cell-like properties of a population of sorted CD44-high/CD24-low cells (BCSCs) MCF7-Tet-On-SrcDN cells that can form mammospheres in culture and initiate tumor growth in vivo. They found that induction of SrcDN expression in mammospheres increased the level of pY397 FAK, as reported previously, but that the level of E-cadherin was unaffected. They went on to show that mammosphere formation efficiency was reduced ~40% by the third passage of CD44-high/CD24-low cells when SrcDN was induced, suggesting that Src function is needed for BCSC self-renewal. To define the effect of SrcDN induction on BCSCs, immunoblotting for proteins of interest and a global proteomic analysis was carried on control and SrcDN expressing mammospheres. Decreases in cyclin D1, Nanog, Oct4 and ALDH1 were noted, consistent with reduced stemness. The global proteomic analysis revealed decreased levels of 15 proteins, with proteins involved in glycolysis being prominent. Next they tested whether SrcDN expression affected glycolysis, and found reduced glucose uptake and lactate excretion, paralleling a reduction in GLUT1, HK2 and LDHA levels, and decreased hexokinase, pyruvate kinase and lactate dehydrogenase enzymatic activity, all consistent with a reduced glycolytic rate. They also found that there were reduced levels of MCT1, the lactate transporter, explaining the observed rise in the level of intracellular lactate. Finally, they showed that cells dissociated from SrcDN-expressing mammospheres had altered maximum oxygen consumption rates in cells cultured in pyruvate-containing medium, albeit without a significant effect on ATP levels.

The observed effects of SrcDN expression on the stemness and metabolic properties of the CD44-high/CD24-low population of stem cell-like cells isolated from cultured MCF7-SrcDN cells are quite interesting, and suggest that that Src family kinase activity is needed to maintain the stemness state and glycolytic properties of BCSC cells. The data are largely convincing, and imply an important role for Src famliy kinase signaling in maintaining a stem cell-like population, although no mechanistic insights are presented into how Src kinase activity results in the observed changes in stem cell and metabolic genes.

Points: 1. Figure 1: The CD44-high/CD24-low population of BCSCs was isolated from MCF7-Tet-On-SrcDN cultures without induction of SrcDN, and then cells were plated to form mammospheres. However, it is not clear from the legend how long the mammospheres, once formed, were treated with DOX prior to the immunoblotting analysis or what happened to the acinar structures during the DOX treatment period, e.g., do they maintain their morphology. It is also not clear what happens to the phenotype of BCSC cells once a mammosphere has been formed - do all the cells retain their BCSC phenotype or have some of them differentiated back into CD44-high/CD24-high cells. The authors report that when mammospheres were dissociated into single cells and then re-plated they could form mammospheres, but information about the efficiency of this process was not provided. If the dissociated cells are re-analyzed by FACS, what fraction are CD44-high/CD24-low and what fraction are CD44-high/CD24-high. If the cells are treated with DOX either prior to (i.e. as mammospheres) or during the re-plating process with DOX, how does this affect the efficiency of mammosphere formation (see point 2)?

2. Figure 2A: It is not clear from the legend or text exactly what was done in this experiment. Were the single cells that were plated obtained from mammospheres that had been pretreated with DOX, or were the cells continuously treated with DOX during plating and mammosphere outgrowth? This needs to be spelled out in the legend.

3. Figure 2B: How long was SrcDN induced for in the mammospheres prior to harvesting for the proteomic analyses?

4. Table 1: It is unclear from the text and the supplementary materials how many total proteins were identified through the MS analysis, i.e. the 15 protein decreases and 4 protein increases need to have a denominator of the total number of proteins identified. By comparative immunoblotting of control and DOX-induced MCF7-SrcDN mammospheres, the authors showed that the levels of several proteins involved in stem cell function and the glycolytic pathway were decreased upon SrcDN induction. However, it is not clear how many of these proteins were also found to be downregulated in their proteomic dataset - none of the proteins tested by immunoblotting is listed in Table S1, but I did not check the data deposited in the ProteomeXchange.

5. Figure 2: Have the authors carried out a parallel RNA-seq analysis on control and DOX-induced MCF7-SrcDN cells to determine which of the observed changes in protein levels can be attributed to decreased levels of the cognate mRNAs (such an analysis does not seem to be reported in their prior studies on MCF7-Tet-On-SrcDN cells). It would be particularly interesting to know whether SrcDN expression affects the levels of Myc and HIF1 RNA expression, since these transcription factors are known to promote expression of the HK2 and LDHA genes. Also, are there any changes in the levels of expression of RNAs encoding mitochondrial ETC genes? Likewise, analysis of the differences in RNA expression patterns between CD44-high/CD24-low BCSCs and the CD44-high/CD24-high cells would be informative.

6. Figure 3: Although an ~50% decrease in pY10 LDHA level upon DOX induction of SrcDN is reported in panel D, there is a concern about how accurate this value is - the low level of the pY10 immunoblot signal means that the percent decrease will be very dependent on how accurately the background signal, which needs to be subtracted, was estimated. A bigger decrease than the observed 50% in pY10 LDHA level might have been expected given the very high level of SrcDN expression compared to that of endogenous c-Src in the induced cells. In this regard, it has been reported (ref. 18) that the level of pY10 in MCF7 cells is very low compared to that in other more advanced breast cancer lines.

Minor points: 1. The first paragraph of the results is entitled “SrcDN reduced self-renewal of MCF7-BCSCs” but no such data are described in this section.

2. While many readers may know what MCF7 cells are, the authors should describe them in the Introduction, i.e. they are ER/PR-positive, HER2-negative, luminal A-type, non-metastatic human breast cancer cells.

3. Do the cellular phenotypes of DOX treated BCSCs revert when DOX is withdrawn, and if so how rapidly?

4. Page 12, second paragraph: The authors say that the endogenous c-Src cannot be distinguished from the exogenously expressed SrcDN by proteomic analysis, but this is not actually true, since they expressed chicken c-SrcDN in human cells, and there are a significant number of sequence differences between the two c-Src’s, which would mean that the chicken and human c-Src proteins would generate unique peptides that could be quantified.

5. Although technically more challenging because of the small cell numbers, it would have been interesting to analyze the sorted CD44-high/CD24-low BCSC MCF7 cell population directly for changes in protein levels, glycolysis rates, etc, compared to the CD44-high/CD24-high population.

Reviewer #2: In this study, the authors tried to investigate whether SrcDN reduces tumorigenesis of MCF7 cells via inhibiting breast cancer stem cells. They isolated BCSCs and non-BCSCs based on ESA, CD44, and CD24 markers and evaluated the effect of SrcDN on the metabolic changes. Though somewhat interesting, the data are underdeveloped and not supporting the conclusion. The major flaw is that they didn’t show that the changes in glycolysis and decrease in self-renewal were unique to the BCSCs. Except isolating these different cell populations and analysis of the E-Cad levels, there are no experiments done comparing these two groups. Without such analysis, the conclusion of the SrcDN on BSCS to inhibit tumorigenesis is not supported. There were several syntax/grammatical errors throughout the manuscript, which should be improved.

In addition, they should have performed direct analysis to address whether the SrcDN induced glycolysis changes are directly connected to self-renewal properties, since they showed it wasn’t related to ATP production or ROS.

Specific comments:

Figure 1:

1. Figure 1A, isotype control for setting up the gate for the FACS should be included.

2. Figure 1B, it is unclear the significance of this figure, since there is no comparison of spheres formed by CD24- and CD24+ groups. There is lack of scale bars.

3. Other methods have been used to isolate putative BCSCs, particularly using ALDH activity for the luminal breast cancers. To strengthen the claim on BCSCs, the authors should also consider using different ways to evaluate BCSCs.

Figure 2:

1. Figure 2A, possibly due to sample size or heterogeneity in the mammospheres, without seeing the full error bars and with the statistical significance being relatively large, this data is not very strong. It is also unclear what the bar and the error bar show. mean or median? SD or SEM?

2. In Figure 2B, the quantitation for the Nanog and Oct3/4 Western blots were the same, but the blot for Oct3/4 was far less apparent compared to Nanog, especially if there were compared to the same β-Actin loading control.

Figure 3:

1. A-C. As in Figure 2, the lack of full error bars and small sample size makes the data less convincing, especially because each graph has the same inferred statistical significance from very different measurements. Again, it is unclear what the error bars represent.

2. In Figure 3D, The blots for Glut-1 and pY10-LDHA, and LDHA are not as well defined as the others, although it could be due to the antibodies. The quantitative decreases in Glut-1, HK2, and pY10-LDHA/LDHA are not very significant and it is unclear whether such level of a decrease can cause changes in glycolytic activity/output. There are no functional validation or rescue experiments done to link the protein level changes to the glycolysis changes.

6. PLOS authors have the option to publish the peer review history of their article (what does this mean?). If published, this will include your full peer review and any attached files.

Reviewer #1: No

Reviewer #2: No

---

## [Author Response · Author response to Decision Letter 0]

9 Jun 2020

We have elaborated a large document that we named "Reviewers_Reply_03-06-2020" in PDF format where we have answered the questions raised by the Reviewers and the Editor. We must say that we are thankful to the comments raised, as we think that they help us to improve the quality of the manuscript.

---

## [Decision Letter · Decision Letter 1]

24 Jun 2020

c-Src functionality controls self-renewal and glucose metabolism in MCF7 breast cancer stem cells

PONE-D-20-06055R1

Dear Dr. Martín-Pérez,

We’re pleased to inform you that your manuscript has been judged scientifically suitable for publication and will be formally accepted for publication once it meets all outstanding technical requirements.

Kind regards,

Antimo Migliaccio, M.D.

Academic Editor

PLOS ONE

Additional Editor Comments (optional):

Reviewers' comments:

Reviewer's Responses to Questions

**Comments to the Author**

1. If the authors have adequately addressed your comments raised in a previous round of review and you feel that this manuscript is now acceptable for publication, you may indicate that here to bypass the “Comments to the Author” section, enter your conflict of interest statement in the “Confidential to Editor” section, and submit your "Accept" recommendation.

Reviewer #1: (No Response)

Reviewer #2: All comments have been addressed

2. Is the manuscript technically sound, and do the data support the conclusions?

Reviewer #1: Yes

Reviewer #2: Yes

3. Has the statistical analysis been performed appropriately and rigorously? 

Reviewer #1: Yes

Reviewer #2: Yes

4. Have the authors made all data underlying the findings in their manuscript fully available?

Reviewer #1: Yes

Reviewer #2: Yes

5. Is the manuscript presented in an intelligible fashion and written in standard English?

Reviewer #1: Yes

Reviewer #2: Yes

6. Review Comments to the Author

Reviewer #1: The authors have satisfactorily addressed most of my comments, and have added some new data as supplementary figures, including Figure S3, where they showed that DOX treatment of MCF7-Tet-On-SrcDN cell-derived mammospheres to induce DN-cSrc expression resulted in a strong decrease in the numbers of CD44-high/CD24-low stem cell-like cells, which have high tumorigenicity, and a corresponding increase in the number of CD44-high/CD24-high cells, which show reduced tumorigenicity, consistent with the observed DOX-induced decrease in stem cell factors such as Nanog.

Reviewer #2: (No Response)

7. PLOS authors have the option to publish the peer review history of their article (what does this mean?). If published, this will include your full peer review and any attached files.

Reviewer #1: Yes: Tony Hunter

Reviewer #2: No

---

## [Editor Report · Acceptance letter]

30 Jun 2020

PONE-D-20-06055R1 

c-Src functionality controls self-renewal and glucose metabolism in MCF7 breast cancer stem cells 

Dear Dr. Martín-Pérez:

I'm pleased to inform you that your manuscript has been deemed suitable for publication in PLOS ONE. Congratulations! Your manuscript is now with our production department. 

Kind regards, 

on behalf of

Dr. Antimo Migliaccio 

Academic Editor

PLOS ONE